## [Decision Letter · Decision Letter 0]

13 Mar 2026

PONE-D-26-04379Implicit bias in safety-aligned large language models: A multi-faceted evaluation of clinical decision-making and health equityPLOS One

Dear Dr. Yu,

Thank you for submitting your manuscript to PLOS ONE. After careful consideration, we feel that it has merit but does not fully meet PLOS ONE’s publication criteria as it currently stands. Therefore, we invite you to submit a revised version of the manuscript that addresses the points raised during the review process.

We look forward to receiving your revised manuscript.

Kind regards,

Thiago P. Fernandes, PhD

Academic Editor

PLOS One

**Journal Requirements:**

“This work was supported by the Chengdu University of Traditional Chinese Medicine in 2025.”

5. We note that there is identifying data in the Supporting Information file < MIAT_Code_and_Data.zip >. Due to the inclusion of these potentially identifying data, we have removed this file from your file inventory. Prior to sharing human research participant data, authors should consult with an ethics committee to ensure data are shared in accordance with participant consent and all applicable local laws.

-Location data

Please remove or anonymize all personal information, ensure that the data shared are in accordance with participant consent, and re-upload a fully anonymized data set. Please note that spreadsheet columns with personal information must be removed and not hidden as all hidden columns will appear in the published file.

**Additional Editor Comments:**

Please respond to all comments and highlight the changes in the revised manuscript.

Reviewers' comments:

Reviewer's Responses to Questions

**Comments to the Author**

1. Is the manuscript technically sound, and do the data support the conclusions?

Reviewer #1: Yes

Reviewer #2: Yes

Reviewer #3: Yes

2. Has the statistical analysis been performed appropriately and rigorously?

Reviewer #1: Yes

Reviewer #2: Yes

Reviewer #3: Yes

3. Have the authors made all data underlying the findings in their manuscript fully available?

Reviewer #1: Yes

Reviewer #2: Yes

Reviewer #3: Yes

4. Is the manuscript presented in an intelligible fashion and written in standard English?

Reviewer #1: Yes

Reviewer #2: Yes

Reviewer #3: Yes

5. Review Comments to the Author

Reviewer #1: 1. The application of GAI in medicine and nursing is a serious issue.

2. Currently, there is extreme caution globally regarding the use of GAI in medicine due to medicine has a very low tolerance for error, even allowing for minor mistakes.

3. The paper stated that professionals must adopt an "AI-aware" stance. They should critically evaluate the algorithm's output, treating it as a fallible "second opinion," not objective truth, ensuring that human judgment remains the ultimate guarantee of fair patient care. This is a very good foundation. Without this layer of safety, even the slightest mishap could cause humanity to lose faith in AI-assisted healthcare.

4. The paper's prioritization of GAI algorithms for debiasing has made a significant contribution to medicine.

5. It's necessary to consider the possibility of major mistakes arising from accumulated facts inferring, while simultaneously relying on artificial intelligence at various levels to remove trivial details in routine tasks.

6. This paper primarily discusses GAI bias in word usage across different models. Besides AI vigilance, a process of decision attribution should be established within the system so that when GAI commits a systemic error, humans can quickly eliminate a series of systemic errors.

Reviewer #2: This manuscript employs technical methods to assess the potential implicit biases of various mainstream LLMs in the medical field and their impact on clinical decision-making and health equity. It ingeniously adapts the traditional IAT into test frameworks suitable for large models, demonstrating high innovation.

Methodologically, when the author verified whether implicit associations predict discriminatory decisions, they only conducted a paired-prompt analysis on the DeepSeek-Chat model. This weakens the generalizability of the conclusion. It is suggested to supplement the comparison results of at least one closed-source advanced model , such as GPT-4 et al to demonstrate that the finding that "implicit associations predict discriminatory decisions" is not confined to a single model architecture.

I couldn't find supplementary materials to verify the test data of this research. It is suggested that specific examples of the construction of the Prompt in both Chinese and English be provided in the main text to demonstrate what kind of medical scenarios the "situational decision-making" that the model faces in the Relative Decision Test is. This would help clinical physician readers understand the mechanism of AI bias more intuitively.

This study did not employ external validation data and did not verify the issue of lexical association bias in the real world. Although the authors raised this issue in the limitations, it is obvious that the same problem was also raised in reference 5. Therefore, as a new study, efforts should be made to address this issue rather than allowing such an obvious defect to persist.

The statement mentioned that the data is in the Supplementary. It is necessary to ensure that the code repository containing all the prompt templates and API call parameters, such as the GitHub repository link is fully uploaded before submission to guarantee the reproducibility of the research.

Reviewer #3: 1. The LLMs examined in this study were trained on explicit, digitally accessible data, including text and images. However, implicit bias is inherently difficult to measure even within traditional psychological research, and this challenge is further compounded in the context of LLMs. These models undergo alignment training that systematically orients their outputs toward socially desirable responses, making it methodologically difficult to bypass this framework. This constitutes an inherent limitation of the current study's design.

2. Traditional implicit bias measures, such as the IAT, rely on response latency as a core indicator, premised on the assumption that conceptually proximate pairings elicit shorter reaction times. The author is suggested to provide supplementary analysis on whether measurable differences in computational response time were observed across models or the underlying distances between the concepts in statistical spaces. If no significant differences were found, the authors should address possible explanations. For instance, that model outputs are fundamentally a function of the current prompt and context, and that computational latency reflects architectural properties rather than cognitive distance. The validity of response time as an index of bias in this context therefore warrants critical discussion.

3. The experimental design includes relatively few scenarios grounded in medical or healthcare-specific contexts. The overall framework resembles a series of forced-choice tasks rather than simulations of authentic clinical decision-making, which may limit the ecological validity of the findings.

4. It is unclear whether all prompts and tasks in this study were administered within the same agent session (i.e., a single conversation thread). If so, the authors should address the potential for context reinforcement across turns, whereby earlier responses may systematically influence subsequent outputs. Additionally, it is worth considering whether the models themselves were pre-trained on literature related to implicit bias measurement. If so, the models may have been capable of inferring the intent of the assessment, potentially confounding the results.

5. In the Relative Decision Task (p.6), the Decision Bias Score is operationalized on a scale of 0 to 1, a design that obscures the directionality of bias. Furthermore, treating bias as a binary construct oversimplifies the phenomenon and fails to account for intersectionality. For example, the compounded effects of race and gender may produce bias patterns that are invisible when each dimension is examined in isolation.

6. In the Paired Prompt Analysis, the bias priming of certain prompts is relatively transparent, which risks activating the models' safety filters. In such cases, the outputs may reflect the operation of safety mechanisms rather than the models' underlying bias tendencies, potentially producing false negatives in bias detection. Furthermore, all experimental materials were administered in English, despite the inclusion of culturally sensitive content involving race and comparisons between Western and traditional Chinese medicine. Given that the models tested were developed by companies operating across different linguistic and cultural contexts, it remains an open question whether language and cultural framing systematically influence the nature and direction of the biases elicited.

7. This study makes a novel contribution by translating implicit bias measurement concepts from human psychology to the evaluation of LLMs. First, in classical psychological experiments, the instruction set and testing environment are critical elements for ensuring measurement validity; it should be clarified whether these elements were systematically incorporated into the prompt design. Second, the inclusion of responses from healthcare professionals as a reference benchmark would allow for a meaningful comparison between model outputs and human clinical judgment, and would more directly speak to the stated focus of the paper.

8. Gender and racial implicit bias emerge as central findings in the category-level analysis. However, it is important to consider whether these observed differences may partly reflect established clinical or epidemiological disparities already documented within the medical literature. If so, some portion of the model's outputs may be reproducing statistical patterns present in the training data rather than exhibiting bias in a normative sense. Distinguishing between outputs that mirror real-world variation and those that represent bias warranting correction is a fundamental interpretive challenge that the authors should address explicitly.

6. PLOS authors have the option to publish the peer review history of their article (what does this mean?). If published, this will include your full peer review and any attached files.

Reviewer #1: **Yes:** Zih-Ping Ho

Reviewer #2: **Yes:** Hanqing Zhao

Reviewer #3: **Yes:** Yi-Ju Lee

---

## [Author Response · Author response to Decision Letter 1]

1 Apr 2026

A detailed point-by-point response to all editor and reviewer comments has been provided in the uploaded file “Response to Reviewers”.

---

## [Decision Letter · Decision Letter 1]

15 Apr 2026

PONE-D-26-04379R1Implicit bias in safety-aligned large language models: A multi-faceted evaluation of clinical decision-making and health equityPLOS One

Dear Dr. Yu,

Thank you for submitting your manuscript to PLOS ONE. After careful consideration, we feel that it has merit but does not fully meet PLOS ONE’s publication criteria as it currently stands. Therefore, we invite you to submit a revised version of the manuscript that addresses the points raised during the review process.

Please respond to all comments and highlight them in the revised ms.

We look forward to receiving your revised manuscript.

Kind regards,

Thiago P. Fernandes, PhD

Academic Editor

PLOS One

Journal Requirements:

Reviewer's Responses to Questions

**Comments to the Author**

1. If the authors have adequately addressed your comments raised in a previous round of review and you feel that this manuscript is now acceptable for publication, you may indicate that here to bypass the “Comments to the Author” section, enter your conflict of interest statement in the “Confidential to Editor” section, and submit your "Accept" recommendation.

Reviewer #2: All comments have been addressed

Reviewer #3: (No Response)

2. Is the manuscript technically sound, and do the data support the conclusions?

Reviewer #2: Yes

Reviewer #3: (No Response)

3. Has the statistical analysis been performed appropriately and rigorously?

Reviewer #2: Yes

Reviewer #3: (No Response)

4. Have the authors made all data underlying the findings in their manuscript fully available?

Reviewer #2: Yes

Reviewer #3: (No Response)

5. Is the manuscript presented in an intelligible fashion and written in standard English?

Reviewer #2: Yes

Reviewer #3: (No Response)

6. Review Comments to the Author

Reviewer #2: The author has made active and systematic revisions in response to the previous round of review comments, supplemented some experiments, and the main issues of the paper have been substantially addressed. However, there are still some areas that need improvement.

1. The author reports that the Debiasing Prompt is significantly effective on GPT-4o (P=0.026), but shows no statistical significance on DeepSeek-V3 (P=0.089), without discussing the possible reasons for this difference.

2. It is suggested to clarify: (1) the complete version number of the model; (2) the time interval of API calls; (3) the settings of key sampling parameters such as temperature.

Reviewer #3: (No Response)

7. PLOS authors have the option to publish the peer review history of their article (what does this mean?). If published, this will include your full peer review and any attached files.

Reviewer #2: **Yes:** Hanqing Zhao

Reviewer #3: **Yes:** Yi-Ju Lee

---

## [Author Response · Author response to Decision Letter 2]

20 Apr 2026

We have provided a point-by-point response to the reviewer comments and revised the manuscript accordingly. All revised files have been uploaded.

---

## [Editor Report · Decision Letter 2]

22 Apr 2026

Implicit bias in safety-aligned large language models: A multi-faceted evaluation of clinical decision-making and health equity

PONE-D-26-04379R2

Dear Dr. Yu,

We’re pleased to inform you that your manuscript has been judged scientifically suitable for publication and will be formally accepted for publication once it meets all outstanding technical requirements.

Kind regards,

Thiago P. Fernandes, PhD

Academic Editor

PLOS One
---

## [Editor Report · Acceptance letter]

PONE-D-26-04379R2

PLOS One

Dear Dr. Yu,

I'm pleased to inform you that your manuscript has been deemed suitable for publication in PLOS One. Congratulations! Your manuscript is now being handed over to our production team.

Kind regards,

on behalf of

Dr. Thiago P. Fernandes

Academic Editor

PLOS One